# RIGNN: A Rationale Perspective for Semi-supervised Open-world Graph Classification

**Xiao Luo**[1]                                                                                                 *xiaoluo@cs.ucla.edu*
**Yusheng Zhao**[2]                                                                              *yusheng.zhao@stu.pku.edu.cn*
**Zhengyang Mao**[2]                                                                       *zhengyang.mao@stu.pku.edu.cn*
**Yifang Qin**[2]                                                                                            *qinyifang@pku.edu.cn*
**Wei Ju**[2]                                                                                                        *juwei@pku.edu.cn*
**Ming Zhang**[2]                                                                                      *mzhang_cs@pku.edu.cn*
**Yizhou Sun**[1]                                                                                              *yzsun@cs.ucla.edu*

[1] *Department of Computer Science, University of California, Los Angeles*
[2] *School of Computer Science, Peking University*

**Reviewed on OpenReview:** *https://openreview.net/forum?id=qcCE4mC2jI*

## Abstract

Graph classification has gained growing attention in the graph machine learning community and a variety of semi-supervised methods have been developed to reduce the high cost of annotation. They usually combine graph neural networks (GNNs) and extensive semi-supervised techniques such as knowledge distillation. However, they adhere to the closed-set assumption that unlabeled graphs all belong to known classes, limiting their applications in the real world. This paper goes further, investigating a practical problem of semi-supervised open-world graph classification where these unlabeled graph samples could come from unseen classes. A novel approach named Rationale-Informed GNN (RIGNN) is proposed, which takes a rationale view to detect components containing the most information related to the label space and classify unlabeled graphs into a known class or an unseen class. In particular, RIGNN contains a relational detector and a feature extractor to produce effective rationale features, which maximize the mutual information with label information and exhibit sufficient disentanglement with non-rationale elements. Furthermore, we construct a graph-of-graph based on geometrical relationships, which gives instructions on enhancing rationale representations. In virtue of effective rationale representations, we can provide accurate and balanced predictions for unlabeled graphs. An extension is also made to accomplish effective open-set graph classification. We verify our proposed methods on four benchmark datasets in various settings and experimental results reveal the effectiveness of our proposed RIGNN compared with state-of-the-art methods.

## 1 Introduction

Recently, graph-structured data has become omnipresent in the real world (Chen et al., 2022b), and graph classification has received extensive attention with applications in various fields such as molecular chemistry and social analysis (Hansen et al., 2015; Ying et al., 2018a; Lee et al., 2019b; Ying et al., 2018b). Graph neural networks (GNNs) have been demonstrated to be efficient and adaptable for this topic due to their strong capability of representation learning (Lu et al., 2019; Schütt et al., 2017; Gilmer et al., 2017). To be specific, every node receives information from its neighboring nodes, which is then aggregated for incremental node embedding updating. A readout operator is adopted to combine all of these node representations into a graph-level representation after a few iterations (Ying et al., 2018b; Lee et al., 2019b). In this fashion,

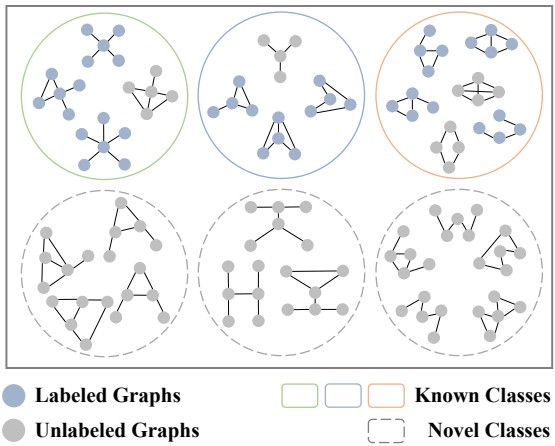

Figure 1: An illustration of our open-world setting. We are required to classify each unlabeled graph example into either one of the known classes or a corresponding novel class.

the learned graph representation is able to reveal the graph structural semantics for effective downstream classification.

Although GNNs have been empirically shown to be effective on numerous benchmarks, they are incredibly data-hungry (Gilmer et al., 2017). Considering that acquiring labels in the real world is usually expensive, one of the dominant solutions is to reduce the labeling cost by semi-supervised learning, which makes use of abundant unlabeled graphs and a limited number of labeled graphs to train GNNs (Sun et al., 2020a; Hao et al., 2020; Yang et al., 2022a; Li et al., 2022a; Yue et al., 2022a). These techniques either use knowledge distillation where a teacher model is imposed to learn generalized graph representations, or pseudo-labeling to annotate unlabeled graphs using their own model. These works almost adhere to the closed-set assumption that unlabeled graphs share the same label space as labeled graphs. Unfortunately, the raw unlabeled set could include samples from unidentified classes in real-world applications. Towards this end, this work generalizes semi-supervised graph classification to a more practical setting called *semi-supervised open-world graph classification*, in which partial unlabeled graphs could belong to unknown classes. In particular, we need to classify each unlabeled graph example into either one of the known classes or a corresponding novel class. Figure 1 provides an example of our problem where colored graphs are with annotations and gray ones are not. Within the same scenario, a similar problem named *semi-supervised open-set graph classification* aims to not only classify samples from known classes correctly but also detect sample graphs from novel classes without further subdivision. We also expect that a generalized algorithm can be easily extended to this similar problem.

The obstacle to semi-supervised open-world graph classification is the broken closed-set assumption. Several studies on open-world recognition primarily concentrate on images and texts (Rizve et al., 2022; Cao et al., 2022; Nayeem Rizve et al., 2022), while our problem on irregular graph data requires us to tackle new challenges as follows: (1) **Complex structured data**. Our problem needs to deal with both attribute-level and structure-level information with varying graph sizes, densities and homophily. Worse yet, the involution of samples from novel classes would further disturb the representation learning of samples from known classes. (2) **The impact of noncrucial components.** The complex data generation procedure may include crucial and noncrucial components and only the former is highly related to target label information. How to extract these informative messages from graphs meanwhile reducing the impact of noncrucial components for effective classification remains an open problem. (3) **Serious label scarcity.** We would encounter serious label scarcity in this problem, especially for novel classes, which could deteriorate the performance of existing semi-supervised GNN-based methods (Yue et al., 2022a). Therefore, an effective strategy to extract approximate semantic information from unlabeled graphs is urgently anticipated.

This work provides a rationale perspective to tackle the problem of semi-supervised open-world graph classification. In particular, we first comprehend the challenges in this problem by understanding the logic of the data generation process where a graph is made up of rationale and non-rationale components. Then, a novel

method named Rationale-Informed GNN (RIGNN) is developed, which integrates rationale learning into effective graph representation learning. To be more precise, we first build a relational detector to select the crucial components and a feature extractor is utilized to extract them into rationale representations. To learn rationale information related to target semantics, we not only maximize the mutual information between rationale representations with the same semantics, but also minimize the mutual information between rationale features and their non-rationale features generated by complementary components. Both contrastive learning and adversarial learning are adopted to implement effective rationale representation learning. In addition, to tackle the label scarcity, we measure the pairwise distance between rationale features and then construct a graph-of-graph based on geometrical relationships, which guides the enhancement of rationale representations. We further add a regularization term to guarantee accurate and balanced predictions for unlabeled graphs. Our work can also be easily extended to accomplish effective open-set graph classification, where outliers only need to be rejected from datasets rather than elaborate classification into different novel classes. We verify our proposed methods on four benchmark datasets in various open-world and open-set settings and experimental results reveal the effectiveness of our proposed RIGNN compared with a variety of state-of-the-art methods. The contribution of this paper can be summarized as follows:

- *New Problem:* We study the problem of semi-supervised open-world graph classification, which breaks the close-set assumption for more generalized flexible real-world applications.

- *Novel Approach:* We develop a novel approach named RIGNN, which involves a rationale perspective in effective graph representation learning. Moreover, a graph-of-graph is constructed to extract the semantic guidance in unlabeled graphs for the enhancement of graph representations.

- *Extensive Experiments:* We verify the effectiveness of our proposed RIGNN by comparing with competitive baselines on four benchmark datasets in various settings.

## 2 Related Work

### 2.1 Graph Classification

Graph neural networks (GNNs) have gained growing attention for graph machine learning problems in recent years (Guo et al., 2022; Zhao et al., 2021; Liu et al., 2021). Graph classification is one of these fundamental problems with extensive applications in computer vision (Jiao et al., 2022), social analysis (Wu et al., 2019) and biology (Xia & Ku, 2021). GNN-based approaches usually follow the message passing mechanism (Ying et al., 2018b; Lee et al., 2019b), which combines structural semantics and node attributes in an iterative fashion. These node representations are then compressed into a graph representation for classification using a pooling procedure. Due to the restricted availability of labels in the real world, semi-supervised graph classification methods have become more popular in research. These approaches use a large number of unlabeled graphs and a few of labeled graphs to maximize the performance of GNNs (Li et al., 2019; Sun et al., 2020a; Hao et al., 2020; You et al., 2020b; Ju et al., 2022; Yang et al., 2022a). However, they do not take into account the situation that the raw graph set could contain samples from unidentified classes. In light of this, we investigate a generalized and practical problem of semi-supervised open-world graph classification.

### 2.2 Rationale Extraction

Rationale extraction or rationale discovery has been incorporated into numerous machine learning applications such as video question answering (Li et al., 2022c), domain generalization (Zhang et al., 2023), sentiment analysis (Yue et al., 2022b), and text classification (Chan et al., 2022). The basic idea of rationale extraction is to extract the crucial part of the input, which facilitates the model performance and explainability. For example, UNIREX explores different Transformer-based rationale extractors to fit multiple priors, which benefit language models in multiple tasks. This topic can also be combined with causality (Zuo et al., 2022) by introducing different intervention techniques (Li et al., 2022b; Wang et al., 2022). Rationale extraction has recently been combined with GNNs to overcome potential out-of-distribution shifts in graph classification (Sui et al., 2022; Yang et al., 2022b). RGCL (Li et al., 2022b) studies the invariant rationale

discovery and then generates augmented graphs from a rationale-aware perspective for effective graph contrastive learning. CIGA (Chen et al., 2022b) describes potential distribution variances on graphs with causal models and extends the invariance principle to graph data. Compared with these methods, our proposed RIGNN learns rationale features based on information theory under label scarcity, which facilitates effective graph classification in both open-world and open-set settings.

### 2.3 Open-set and Open-world Recognition

Open-set recognition expects the model to reject instances from new classes while taking into account the inductive learning configuration (Sun et al., 2020b; Zhou et al., 2021; Kong & Ramanan, 2021; Xia et al., 2021). Open-world recognition further requires us to separate these rejected instances based on their semantics (Rizve et al., 2022; Cao et al., 2022; Nayeem Rizve et al., 2022). Existing open-set and open-world techniques can be divided into generating and discriminative models. To match realistic environments, discriminative models often modify the softmax layer utilizing one-vs-rest units (Scheirer et al., 2012), calibration (Scheirer et al., 2014) and optimal transport (Rizve et al., 2022). In contrast, generative models use conditional auto-encoders (Oza & Patel, 2019) and data augmentation (Ditria et al., 2020) to forecast the distribution of unobserved classes. Recently, self-supervised learning has been incorporated to learn from augmented samples (Rizve et al., 2022). Open-set recognition has been further considered simultaneously with domain shifts (Panareda Busto & Gall, 2017). However, these methods usually focus on Euclidean data, while our RIGNN aims to handle complicated graph data and extract crucial features from a rationale perspective.

## 3 Preliminaries

### 3.1 Problem Definition

A graph is denoted as $\mathcal{G} = (\mathcal{V}, \mathcal{E})$ where $\mathcal{V}$ and $\mathcal{E}$ is the node set and edge set, respectively. $\boldsymbol{X} \in \mathbb{R}^{|\mathcal{V}| \times F}$ denotes the node attribute matrix with the attribute dimension $F$ and $\boldsymbol{A} \in \mathbb{R}^{n \times n}$ denotes the adjacent matrix. In the setting of semi-supervised open-world graph classification, we have a dataset $\mathcal{D}$, which includes a labeled subset $\mathcal{D}^l = \{\mathcal{G}_1, \mathcal{G}_2, \cdots, \mathcal{G}_{N^l}\}$ containing $N^l$ labeled samples and an unlabeled subset $\mathcal{D}^u = \{\mathcal{G}_{N^l+1}, \mathcal{G}_{N^l+2}, \cdots, \mathcal{G}_{N^l+N^u}\}$ containing $N^u$ unlabeled samples. The class set of labeled data and the whole data is denoted as $\mathcal{C}^l$ and $\mathcal{C}$. Closed-world semi-supervised classification implies $\mathcal{C}^l = \mathcal{C}$ while in our settings we have $\mathcal{C}^l \subset \mathcal{C}$, and $\mathcal{C}^u = \mathcal{C} \backslash \mathcal{C}^l$ contains novel classes. We aim to learn a model, which classifies unlabeled graphs from both known and novel classes into their corresponding classes in $\mathcal{C}$.

### 3.2 Graph Neural Networks

We provide a brief overview of graph neural networks (Kipf & Welling, 2017; Xu et al., 2019), which are mainstream techniques for encoding graph-structured data. They often adopt the neighborhood aggregation strategy to extract structural data. In particular, the updating rule for each node $i \in \mathcal{G}$ at layer $l$ is written as follows:

$$
\begin{aligned}
\boldsymbol{n}_i^{(l)} &= \text{AGGREGATE}^{(l)} \left( \left\{ \boldsymbol{v}_j^{(l-1)} : j \in \mathcal{N}(i) \right\} \right), \\
\boldsymbol{v}_i^{(l)} &= \text{COMBINE}^{(l)} \left( \boldsymbol{v}_i^{(k-1)}, \boldsymbol{n}_i^{(l)} \right),
\end{aligned}
\tag{1}
$$

where $\mathcal{N}(i)$ collects the neighboring nodes around $i$. $\boldsymbol{v}_i^{(l)}$ and $\boldsymbol{n}_i^{(l)}$ denote the node representation and the neighborhood representation at layer $l$. $\text{AGGREGATE}^{(l)}(\cdot)$ and $\text{COMBINE}^{(l)}(\cdot)$ denote the aggregation and combination operators at layer $l$, respectively. After stacking $L$ layers, a readout operation is adopted to summarize all these node representations at the final layer into a graph-level representation $\boldsymbol{z} \in \mathbb{R}^d$ where $d$ is the hidden dimension. In formulation,

$$
\boldsymbol{z} = \text{READOUT} \left( \left\{ \boldsymbol{v}_i^{(L)} \right\}_{i \in \mathcal{V}} \right),
\tag{2}
$$

where $\text{READOUT}(\cdot)$ could be represents averaging or complicated pooling procedures (Ying et al., 2018b; Lee et al., 2019b).

### 3.3 Graph Contrastive Learning

We briefly introduce the framework of graph contrastive learning for unsupervised graph representation learning (You et al., 2020b; 2021). Typically, these methods usually maximize the mutual information between input graphs and their representations by comparing the similarity between two augmented views of each input with the similarity between different samples. Given a dataset $\{\mathcal{G}_i\}_{i=1}^N$ and a stochastic augmentation operator $\mathcal{T}(\cdot)$, they first construct positive pairs as $\{\mathcal{G}_i^{(1)}, \mathcal{G}_i^{(2)}\}_{i=1}^N$ with $\mathcal{G}_i^{(r)} = \mathcal{T}(\mathcal{G}_i)$. Then a graph encoder $g(\cdot)$ transfer augmented graphs into representations, i.e., $\boldsymbol{z}_i^{(r)} = g(\mathcal{G}_i^{(r)})$. Given a batch $\mathcal{B}$ and a temperature parameter $\tau$, the normalized temperature-scaled cross entropy (NT-XENT) loss is used to conduct contrastive learning:

$$\mathcal{L} = \frac{1}{|B|} \sum_{\mathcal{G}_i \in \mathcal{B}} -\log \frac{exp(\boldsymbol{z}_i^{(1)} \star \boldsymbol{z}_i^{(2)}/\tau)}{\sum_{\mathcal{G}_{i'} \in \mathcal{B}} exp(\boldsymbol{z}_i^{(1)} \star \boldsymbol{z}_{i'}^{(2)}/\tau)}, \tag{3}$$

where $\star$ denotes the cosine similarity between two vectors and $\mathbb{E}$ calculates the empirical average over a dataset since every sample has the same probability to be sampled.

## 4 Methodology

### 4.1 Overview

This paper studies the problem of semi-supervised open-world graph classification. Although a variety of methods have been put forward to address the label scarify in graph classification (Li et al., 2019; Sun et al., 2020a; Hao et al., 2020; You et al., 2020b; Yang et al., 2022a; Xie et al., 2022), they usually adhere to the close-world assumption that unlabeled graphs belong to known classes. This assumption restricts their applications in the real world.

Here, we propose a novel method named rationale-informed graph neural network (RIGNN) to solve this problem. The basic idea is to discover the rationale elements for effective graph representation learning. In particular, we first comprehend the challenges in this problem, and then incorporate rationale discovery into graph representation learning based on information theory, which retains components related to semantics labels. Moreover, we construct a graph-of-graph, which detects semantic proximity in unlabeled graphs to enhance our rationale-informed representation learning. Finally, we summarize our semi-supervised open-world learning framework and make an extension. More details can be seen in Figure 2.

### 4.2 A Rationale View for Graph Generation

We first comprehend the challenges in this problem by illustrating the graph generation process. To begin, a graph $\mathcal{G}$ is constructed using both rationale and non-rationale components, i.e., $R$ and $NR$. Here $R$ is closely tied to intrinsic property which is highly relevant to our downstream classification. $NR$ refers to the part irrelevant to our target property, which can be varied due to different backgrounds. However, previous graph classification approaches feed both $R$ and $NR$ as a whole into a message passing neural network, which clearly suffers from the influence of $NR$. Therefore, we expect to reduce the impact of $NR$ to generate discriminative rationale representations, which is helpful to generate confident and accurate predictions $Y$ even with $Y \in \mathcal{C}^u$.

Then, we attempt to incorporate the logic in graph representation learning.

### 4.3 Representation learning via Rationale Extraction

To perform effective open-world classification, we need to be more cautious when generating rationale features to get rid of the impact of the non-rationale part. Since the data generation process cannot be intervened, we turn to information theory instead to learn invariant representation under varying non-rationale components. To achieve this, we introduce a relational detector to generate the probability of each node carrying rationale information. We train the relational detector along with a feature extractor, which produces rationale features

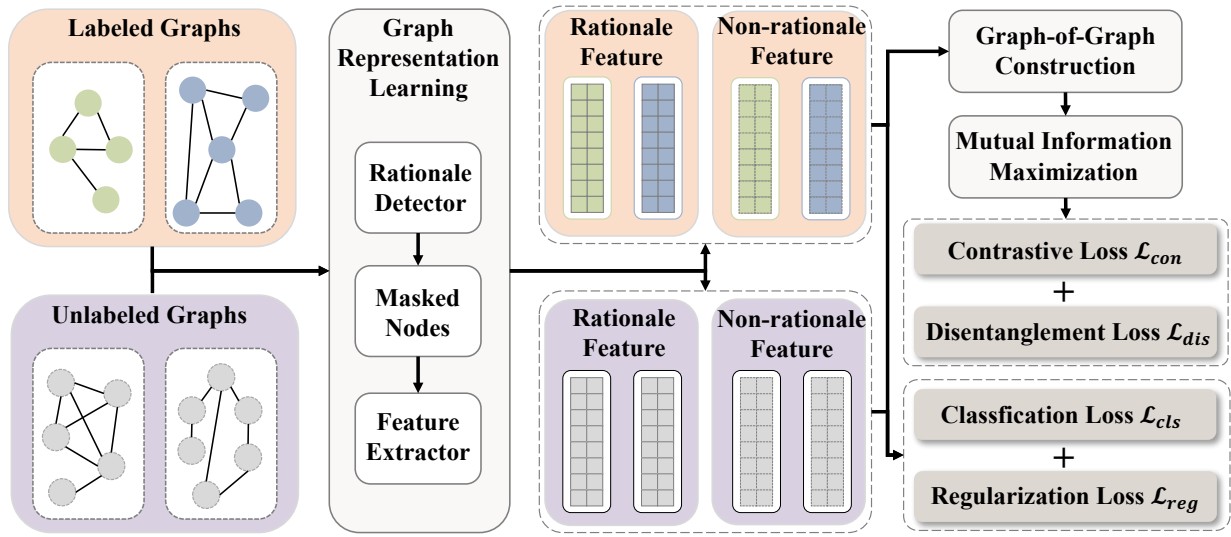

Figure 2: Illustration of the proposed framework RIGNN. Our RIGNN utilizes a relational detector and feature extractor to generate rationale features related to semantic labels and complementary non-rationale features. Moreover, we construct a graph-of-graph to extract the additional semantic information in the unlabeled set. The whole model is optimized using the combination of four objectives.

to not only maximize the mutual information with label information, but also disentangle with non-rationale components.

In detail, our relational detector is a message passing neural network $f_\theta(\cdot)$ with parameters $\theta$, which first stacks graph convolution layers to generate hidden representations and then utilize a multi-layer perception (MLP) to generate the probability that each node should be kept. Formally, a mask vector based on importance scores $\boldsymbol{M} \in (0,1)^{|\mathcal{V}| \times 1}$ is defined as:

$$\mathbf{M}_i = f_\theta(i;\mathcal{G})\mathbf{1}_{\{f_\theta(i;\mathcal{G})>\epsilon\}}, \tag{4}$$

where $f_\theta(i;\mathcal{G})$ denotes the corresponding output of $f_\theta(\cdot)$ for node $i$ and we keep $\alpha|\mathcal{V}|$ nodes above a pre-defined threshold $\epsilon$. $\alpha$ is a predefined parameter to decide the proportion of the kept nodes. We add continuous values in the mask for efficient gradient updating. Then, the node attribute after masking would be $\boldsymbol{X}^r = \boldsymbol{X} \odot \mathbf{M}$ which $\odot$ denotes the Hadamard product of two matrices. The removed information in the attribute matrix is $\boldsymbol{X}^{nr} = \boldsymbol{X} \odot (\mathbf{1} - \mathbf{M})$, which indicates non-rationale information. With the rationale attribute matrix and the non-rationale attribute matrix, we can generate rationale feature $\boldsymbol{z}^r$ and non-rationale feature matrix $\boldsymbol{z}^{nr}$ using a feature extractor, which is another message passing neural network $g_\phi(\cdot)$ as follows:

$$\boldsymbol{z}^r = g_\phi(\mathcal{G}; \boldsymbol{X}^r), \tag{5}$$

$$\boldsymbol{z}^{nr} = g_\phi(\mathcal{G}; \boldsymbol{X}^{nr}). \tag{6}$$

To relieve the impact caused by the non-rationale part, we aim to maximize the mutual information between $\boldsymbol{z}^c$ and its label $\boldsymbol{y}$ while minimizing the mutual information between $\boldsymbol{z}^c$ and $\boldsymbol{z}^{nc}$ for disentanglement. In formulation, the objective is:

$$\max_{\phi,\theta} I(\boldsymbol{z}^r; \boldsymbol{y}) - I(\boldsymbol{z}^r; \boldsymbol{z}^{nr}). \tag{7}$$

However, label information is unavailable in $\mathcal{D}^u$. To tackle this, we consider two different graph samples with an identical rationale part, i.e., $\mathcal{G}$ and $\tilde{\mathcal{G}}$. From our previous analysis, their labels, i.e., $\boldsymbol{y}$ and $\tilde{\boldsymbol{y}}$ should be the same, and thus the mutual information between their rationale features should be maximized. In turn, maximizing the mutual information between rationale features with the same label would naturally result in invariant features (Bachman et al., 2019). In other words, when the labels of two graphs are the same, they

should share the same rationale components which are highly relevant to our downstream classification (by maximizing $I(\boldsymbol{z}^c; y)$). Therefore, we revise Equation 7 as follows:

$$\max_{\phi,\theta} I\left(\boldsymbol{z}_Y^r; \tilde{\boldsymbol{z}}_Y^r \mid Y\right) - I(\boldsymbol{z}^r; \boldsymbol{z}^{nr}), \tag{8}$$

where $\boldsymbol{z}_Y^r$ and $\tilde{\boldsymbol{z}}_Y^r$ corresponds to the feature representation of $G$ and $\tilde{G}$ given the same label $Y$.

To maximize $I\left(\boldsymbol{z}_Y^r; \tilde{\boldsymbol{z}}_Y^r \mid Y\right)$, we turn to graph contrastive learning (You et al., 2020b; 2021), which constructs rationale-informed positive pairs (i.e., with the same label $Y$) from two sources. On the one hand, we consider sample pairs with the same labels as positives in $\mathcal{D}^l$. On the other hand, we take each original sample in $\mathcal{D}^u$ and its subgraphs (with a dropout ratio 0.2 (You et al., 2020b)) as positives since we do not have access to the label information. In formulation, we define the positive set as $\mathcal{P} = \{(i,j)|\boldsymbol{y}_i = \boldsymbol{y}_j, \mathcal{G}_i, \mathcal{G}_j \in \mathcal{D}^l\}$ and have the contrastive loss as:

$$\mathcal{L}_{con} = -\frac{1}{|\mathcal{P}|} \sum_{\mathcal{G}_i,\mathcal{G}_j \in \mathcal{D}^l,(i,j)\in\mathcal{P}} \log \frac{e^{\boldsymbol{z}_i \star \boldsymbol{z}_j/\tau}}{\sum_{\mathcal{G}_{j'}\in\mathcal{D}} e^{\boldsymbol{z}_i \star \boldsymbol{z}_{j'}/\tau}} - \frac{1}{|\mathcal{D}^u|} \sum_{\mathcal{G}_i\in\mathcal{D}^u} \log \frac{e^{\boldsymbol{z}_i \star \hat{\boldsymbol{z}}_i/\tau}}{\sum_{\mathcal{G}_{j'}\in\mathcal{D}} e^{\boldsymbol{z}_i \star \boldsymbol{z}_{j'}/\tau}}, \tag{9}$$

where $\tau$ is a temperature parameter set to 0.5 following previous works (You et al., 2020b; Ju et al., 2022) and $\hat{\boldsymbol{z}}_i$ denotes the rationale feature for the subgraph of $\mathcal{G}_i$. Here two positive sets together allow our model to effectively leverage unlabeled data to learn more powerful embeddings. Finally, to minimize $I(\boldsymbol{z}^r, \boldsymbol{z}^{nr})$ for sufficient disentanglement of rationale and non-rationale elements, we build a Jensen-Shannon mutual information estimator $T_\gamma$ (Sun et al., 2020a), which is trained in an adversarial manner. In formulation, we have:

$$\mathcal{L}_{dis} = \frac{1}{|\mathcal{D}|} \sum_{\mathcal{G}_i\in\mathcal{D}} sp(-T_\gamma(\boldsymbol{z}_i^r, \boldsymbol{z}_i^{nr})) + \frac{1}{|\mathcal{D}|^2} \sum_{\mathcal{G}_i,\mathcal{G}_j\in\mathcal{D}} -sp(-T_\gamma(\boldsymbol{z}_i^r, \boldsymbol{z}_j^{nr})), \tag{10}$$

where $sp(\boldsymbol{x}) = \log(1+e^{\boldsymbol{x}})$ is the softplus function. In summary, our model is optimized in a minimax game,

$$\min_{\theta,\phi} \max_{\gamma} \mathcal{L}_{con} + \mathcal{L}_{dis}, \tag{11}$$

To resolve Equation 18, we minimize two sub-objectives till the convergence as follows:

$$\begin{cases} \min_{\theta,\phi} \mathcal{L}_{con} + \mathcal{L}_{dis} \\ \min_{\gamma} -\mathcal{L}_{dis}. \end{cases} \tag{12}$$

From Equation 12, on the one hand, we train the estimator for accurate measurement of mutual information. On the other hand, we update the network parameters to obtain discriminative rationale features satisfying Equation 8.

## 4.4 Representation Enhancement via Graph-of-Graph

We have created rationale features which are highly related to semantic labels. Intuitively, the geometrically nearest neighbors based on rationale features can be considered as semantic-similar graph pairs (Chen et al., 2022a). To make use of abundant unlabeled graphs, a graph-of-graph is constructed to connect independent graphs with similar semantics, providing extra semantic proximity to enhance rationale representation learning.

In detail, we compare the rationale features of graph pairs and measure the similarity using the cosine distance:

$$s_{ij} = \boldsymbol{z}_i \star \boldsymbol{z}_j. \tag{13}$$

Then, we identify k-nearest neighbors (kNNs) of labeled samples to add edges between graph samples where $k$ denotes the number of neighbors. However, due to the label scarify of novel classes, kNNs could introduce false positives by connecting samples from novel classes to the other classes. To handle this, we filter false positives by identifying mutual nearest neighbors (MNN) for unlabeled samples. In other words, we connect

$\mathcal{G}_i$ and $\mathcal{G}_j$ when $\boldsymbol{z}_i \in \text{kNN}(\boldsymbol{z}_j) \wedge \boldsymbol{z}_j \in \text{kNN}(\boldsymbol{z}_i)$ (i.e., $\boldsymbol{z}_i \in \text{MNN}(\boldsymbol{z}_j)$). Therefore, the adjacency matrix of the graph-of-graph is defined as:

$$\mathcal{A}_{ij} = \begin{cases} 1, \boldsymbol{z}_j^r \in \text{kNN}(\boldsymbol{z}_i^r), \mathcal{G}_i \in \mathcal{D}^l \bigvee \boldsymbol{z}_j^r \in \text{MNN}(\boldsymbol{z}_i^r), \mathcal{G}_i \in \mathcal{D}^u \\ 0, \text{otherwise} \end{cases}. \tag{14}$$

Afterward, we view connected graph pairs in the graph-of-graph as positives and add them into the positive set $\mathcal{P}$. In formulation,

$$\mathcal{P} \leftarrow \mathcal{P} \cup \{(i,j) | \mathcal{A}_{ij} = 1\}. \tag{15}$$

Equation 15 enlarges the positive set, which enhances rationale graph representations with the additional guidance of semantic proximity under serious label scarcity.

### 4.5 Framework Summarization

Finally, we incorporate our rationale representations into open-world graph classification. To build a mapping from rationale representations to label space, we add a classifier $h_\phi(\cdot) : \mathbb{R}^d \to \mathcal{R}^{|\mathcal{C}|}$ on the top of $g_\phi(\cdot)$ where the first $|\mathcal{C}^l|$ scores are for the unseen classes, while the last $|\mathcal{C}^u|$ scores are for expected novel classes. Then we minimize the standard classification loss for labeled data and minimize the entropy for unlabeled data to generate informative distributions:

$$\mathcal{L}_{cla} = \mathbb{E}_{\mathcal{G}_i \in \mathcal{D}^l} CE(h_\phi(\boldsymbol{z}_i), \boldsymbol{y}_i)) + \mathbb{E}_{\mathcal{G}_i \in \mathcal{D}^u} H(h_\phi(\boldsymbol{z}_i))), \tag{16}$$

where $CE(\cdot)$ denote the standard cross-entropy loss and $H(\cdot)$ measures the entropy of the distribution. However, minimizing the entropy of predictions for unlabeled graphs could generate trivial solutions which assign the majority of novel samples into a single class (Huang et al., 2020). To tackle this, we introduce a regularization term which minimizes the negative entropy of averaged distributions across the whole dataset:

$$\mathcal{L}_{reg} = \log(|\mathcal{C}|) - H(\boldsymbol{p}), \text{ with } \boldsymbol{p} = \left[p_1, p_2, \cdots, p_{|\mathcal{C}|}\right], \tag{17}$$

where $\boldsymbol{p}[c] = \dfrac{\sum_{\mathcal{G}_i \in \mathcal{D}} h_\phi(\boldsymbol{z}_i)[c]}{\sum_{c'=1}^{|\mathcal{C}|} \sum_{\mathcal{G}_i \in \mathcal{D}} h_\phi(\boldsymbol{z}_i)[c']}$ denotes the summarized probability of belonging to class $c$ in the whole dataset and $\log|\mathcal{C}|$ can make the loss non-negative. In a nutshell, our final objective can be written as follows:

$$\min_{\theta, \phi} \max_\gamma \mathcal{L}_{cla} + \mathcal{L}_{dis} + \mathcal{L}_{reg} + \lambda \mathcal{L}_{con}, \tag{18}$$

where $\lambda$ is a parameter to balance these losses. Similarly, adversarial learning is implemented using the gradient reverse layer (Zhang et al., 2018) to optimize the whole framework as in Equation 12. In practice, we adopt mini-batch stochastic gradient descent to update the whole framework and update the graph-of-graph every cycle, and the total cycle number is $T$. The detailed algorithm is shown in Algorithm 1.

**Complexity.** The computational complexity of our RIGNN mainly depends on the relational detector and the feature extractor. Given a graph $\mathcal{G}$ with the number of nonzeros in the adjacency matrix denoted as $||\boldsymbol{A}||_0$. Recall that $d$ denotes the feature dimension. $L_r$ and $L_f$ denotes the layer number of $f_\theta(\cdot)$ and $g_\phi(\cdot)$, respectively. $|\mathcal{V}|$ is the number of nodes. Obtaining rationale features and non-rationale features takes $\mathcal{O}((L_r + L_f)||\boldsymbol{A}||_0 d + (L_r + L_f)|\mathcal{V}|d^2)$ computational time. From the results, the complexity of the proposed RIGNN is linearly related to $|\mathcal{V}|$, $||\boldsymbol{A}||_0$ and $L_r + L_f$.

### 4.6 Extension to Open-set Graph Classification

Although RIGNN is originally designed for semi-supervised open-world graph classification, it can be extended to open-set graph classification (Luo et al., 2023), which only needs to detect outliers in the unlabeled set. Here, we would adjust the classifier into $\tilde{h}_\phi(\cdot) : \mathbb{R}^d \to \mathcal{R}^{|\mathcal{C}^l|}$ and detect outliers by selecting samples with small confidence scores, i.e., $q = \max_k \tilde{h}_\phi(\mathcal{G})[k]$. Moreover, we will delete the regularization loss $\mathcal{L}_{reg}$ since trivial solutions could not occur. Due to the existence of outliers, the classification loss is limited to

labeled samples and unlabeled samples with high confidence. In formulation, we set a threshold $\mu$ and the set of outliers is $\{\mathcal{G}_i : q_i \leq \mu\}$. The classification is modified into the following equation:

$$\tilde{\mathcal{L}}_{cla} = \mathbb{E}_{\mathcal{G}_i \in \mathcal{D}^l} CE(h_\phi(\boldsymbol{z}_i), \boldsymbol{y}_i)) + \mathbb{E}_{\mathcal{G}_i \in \mathcal{D}^u} \mathbf{1}_{q_i > \mu} H(h_\phi(\boldsymbol{z}_i))). \tag{19}$$

The final objective is modified into:

$$\min_{\theta, \phi} \max_{\gamma} \tilde{\mathcal{L}}_{cla} + \mathcal{L}_{dis} + \lambda \mathcal{L}_{con}. \tag{20}$$

We will also utilize adversarial training for the disentanglement.

## 5 Experiments

In this section, we conduct various experiments on six datasets to validate the effectiveness of our RIGNN. The experimental results show the superiority of RIGNN in both open-world and open-set graph classification settings. Specifically, we will focus on the following research questions (RQs): (1) *RQ1*: What is the performance of our RIGNN compared to baselines in the *open-world* graph classification task? (2) *RQ2*: What is the prediction accuracy of RIGNN compared to baseline models in the *open-set* graph classification task? (3) *RQ3*: What is the influence of rationale representation learning, contrastive learning and graph-of-graph representation enhancement in the model's performance? (4) *RQ4*: Are there any visualization results of the rationale representation learning?

### 5.1 Experimental Setup

**Datasets and Evaluation Protocols.** We utilize four public benchmark graph datasets, i.e., COIL-DEL, Letter-high, MNIST, CIFAR10, REDDIT and COLORS-3 in our experiments. Their statistics are presented in Table 4. We divide all the classes into known classes and unknown classes with details recorded in Table 4. In both the open-world and open-set semi-supervised settings, partial labels are available for samples from known classes and we cannot get access to the labels of examples from the novel classes. We create two scenarios indicating different labeling ratios and denote them as *Easy* (a higher labeling ratio) and *Hard* (a lower labeling ratio), respectively. In particular, the ratio for *Easy/Hard* problems is 0.8/0.5, 0.4/0.2, 0.03/0.01, 0.07/0.03, 0.7/0.3 and 0.8/0.3 for COIL-DEL, Letter-High, MNIST, CIFAR10, REDDIT and COLORS-3, respectively (Luo et al., 2023). We report the classification accuracy to compare the performance. To be more precise, in the open-world setting, Hungarian algorithm (Kuhn, 1955) is adopted to match these unknown classes and calculate the final prediction accuracy. In the open-set setting, we view all these novel classes as a unified class and when the model gives a correct label for samples from known classes or rejects samples from novel classes, we classify them correctly.

**Baselines.** The proposed RIGNN is compared with a range of competing baselines, including graph neural network methods (GraphSAGE (Hamilton et al., 2017), GIN (Xu et al., 2019), GCN (Kipf & Welling, 2017), ASAP (Ranjan et al., 2020), Edge Pooling (Diehl, 2019), TopK Pooling (Gao & Ji, 2019a) and SAG Pooling (Lee et al., 2019a)) and graph contrastive learning methods (GraphCL (You et al., 2020b), GLA (Yue et al., 2022a) and UGNN (Luo et al., 2023)).

**Implementation Details.** We implement the proposed RIGNN with PyTorch and train all the models with an NVIDIA RTX GPU. As for hyperparameters, we set $k$ in the graph-of-graph construction process to 2. For the weight $\lambda$ in the loss function, we set it to 0.1. Their detailed analysis can be found in Section C. The dimension of all hidden features is set to 128. As for the network architecture, we use a two-layer GraphSAGE (Hamilton et al., 2017) to construct the relational detector $f_\theta$ and a three-layer GIN convolution for the feature extractor $g_\theta$. In the middle of the convolutional layer, we implement graph pooling with TopK Pooling (Gao & Ji, 2019b) as default. For the Jensen-Shannon mutual information estimator $T_\gamma$, we concatenate the two inputs and send the feature to a two-layer MLP. A two-layer MLP is also adopted from the classifier $h_\phi$. For the model training, we train the model for 100 epochs in total and utilize the entire dataset for estimating the mutual information. In terms of optimization, we employ the gradient reversal layer (Ganin & Lempitsky, 2015) to realize the adversarial training of $\gamma$. This approach provides

Table 1: Open-world classification accuracy in COIL-DEL, Letter-high, MNIST and CIFAR10 datasets. Both Easy and Hard scenarios are included, and the proposed RIGNN achieves the best performance.

| Methods | COIL-DEL | | Letter-High | | MNIST | | CIFAR10 | | REDDIT | | COLORS-3 | | Average |
|---|---|---|---|---|---|---|---|---|---|---|---|---|---|
| | Hard | Easy | Hard | Easy | Hard | Easy | Hard | Easy | Hard | Easy | Hard | Easy | |
| GraphSAGE | 35.64 | 41.53 | 52.67 | 55.11 | 19.31 | 40.91 | 29.88 | 31.90 | 28.96 | 27.49 | 25.14 | 25.38 | 34.39 |
| GIN | 50.38 | 56.15 | 46.00 | 48.67 | 41.55 | 53.17 | 33.92 | 35.23 | 26.15 | 27.24 | 22.57 | 21.52 | 38.55 |
| ASAP | 45.00 | 57.56 | 30.00 | 47.11 | 34.40 | 55.40 | 34.90 | 35.93 | 27.03 | 27.16 | 33.33 | 35.00 | 38.57 |
| Edge Pooling | 45.90 | 50.00 | 38.22 | 49.78 | 22.02 | 46.58 | 27.58 | 32.23 | 27.03 | 26.57 | 31.90 | 31.29 | 35.76 |
| TopK Pooling | 35.64 | 37.44 | 35.33 | 48.89 | 21.03 | 37.33 | 32.65 | 31.43 | 29.09 | 29.00 | 30.67 | 31.62 | 33.34 |
| SAG Pooling | 41.28 | 46.54 | 43.78 | 48.89 | 28.90 | 54.00 | 28.94 | 29.81 | 23.85 | 28.46 | 29.71 | 28.19 | 36.03 |
| GraphCL | 48.33 | 53.97 | 44.89 | 48.44 | 35.11 | 56.48 | 33.81 | 35.78 | 26.99 | 27.91 | 34.62 | 34.76 | 40.09 |
| GLA | 51.92 | 55.26 | 44.00 | 48.22 | 42.59 | 55.90 | 33.06 | 35.40 | 29.00 | 29.72 | 35.14 | 34.95 | 41.26 |
| RIGNN (Ours) | 52.69 | 61.03 | 46.67 | 50.44 | 52.92 | 61.06 | 36.37 | 39.57 | 29.34 | 29.46 | 35.86 | 35.95 | 44.28 |
| Improvement | 1.4% | 8.7% | 1.5% | 1.3% | 24.3% | 8.1% | 7.2% | 7.7% | 1.2% | -0.8% | 2.0% | 2.7% | 7.3% |

a mechanism to avoid alternative optimization, and ensure the overall training stability. The parameters $\theta$ and $\phi$ are viewed as an integrated unit during the training process. This design choice effectively mitigates the discrepancies in the update frequencies among different components. Moreover, our training process is divided into two phases. We initially warm up the model with labeled data only, ensuring that the parameters reach a stable state before the introduction of complex interactions. Following this, the model is trained jointly with all the available data. We use the Adam optimizer for its well-known efficiency and effectiveness. In the training, we use Adam (Kingma & Ba, 2015) optimizer and set the batch size to 256, with the learning rate set to 0.001.

## 5.2 The Performance of RIGNN in Open-world Graph Classification (RQ1)

The open-world classification accuracy on the datasets COIL-DEL, Letter-High, MNIST, CIFAR10, REDDIT and COLORS-3 compared to the baseline methods is listed in Table 1. From the results, we obtain the following observations:

- Firstly, the proposed RIGNN obtains a consistent lead in both Hard and Easy scenarios on all four datasets, which demonstrates the superiority of the model. In particular, we attribute the performance gain to two aspects: better representation learning with rationale and the representation enhancement according to the constructed graph-of-graph proximity. Learning with rationale helps the model detect the most essential part of the graph and get rid of the non-rationale part, which contributes to the generalization capability of the model to unknown classes. The constructed graph-of-graph and the corresponding contrastive learning improve the model capability to detect semantic proximity among unlabeled instances and to make the best of unlabeled instances. With the enhancement brought by the graph-of-graph, the model is better at classifying graph instances in unknown classes.

- In addition, we observe that our model achieves more significant improvement on the MNIST and CIFAR10 datasets, which contains more nodes and edges in a graph, compared to the Letter-high dataset, which contains fewer nodes on average. One possible explanation for this is that each node in a small graph plays a more important role in the class-determining process than nodes in a large graph. Large graphs like those in MNIST and CIFAR10 tend to contain more non-rationale parts, for example, the nodes representing the background in MNIST and CIFAR10. Therefore, the proposed rationale representation learning contributes less to the classification of small graphs.

- Moreover, we find that existing semi-supervised graph classification methods fail to provide satisfactory accuracy in the open-world classification task, since they are designed for the closed-world graph classification. In comparison, the proposed RIGNN leverages rationale to discover the most essential part in the graph related to the label space and adopts the graph-of-graph construction to make better use of

Table 2: Open-set classification accuracy in COIL-DEL, Letter-high, MNIST and CIFAR10 datasets. Both Easy and Hard scenarios are included, and the proposed RIGNN achieves the best performance.

| Methods | COIL-DEL | | Letter-High | | MNIST | | CIFAR10 | | REDDIT | | COLORS-3 | | Average |
|---|---|---|---|---|---|---|---|---|---|---|---|---|---|
| | Hard | Easy | Hard | Easy | Hard | Easy | Hard | Easy | Hard | Easy | Hard | Easy | |
| GCN | 22.56 | 33.46 | 32.89 | 50.44 | 20.00 | 37.03 | 30.89 | 36.45 | 31.23 | 33.42 | 31.87 | 38.32 | 33.21 |
| GraphSAGE | 37.69 | 39.74 | 40.89 | 58.22 | 19.56 | 41.97 | 32.94 | 35.14 | 24.43 | 26.19 | 31.31 | 34.00 | 35.17 |
| GIN | 41.15 | 44.49 | 48.89 | 57.11 | 29.72 | 62.55 | 27.18 | 33.32 | 34.71 | 35.83 | 31.09 | 33.53 | 39.96 |
| SAG Pooling | 41.03 | 48.46 | 46.22 | 52.22 | 42.49 | 63.22 | 36.41 | 38.83 | 32.81 | 34.74 | 39.07 | 40.21 | 42.98 |
| GraphCL | 56.28 | 60.64 | 56.22 | 63.56 | 49.81 | 69.97 | 37.27 | 40.98 | 32.71 | 36.02 | 38.17 | 39.94 | 48.46 |
| GLA | 56.54 | 61.03 | 60.22 | 63.11 | 48.30 | 70.45 | 38.34 | 41.18 | 34.12 | 36.80 | 39.03 | 40.89 | 49.17 |
| UGNN | 59.36 | 62.95 | 64.00 | 66.00 | 58.50 | 73.04 | 39.73 | 42.03 | 37.73 | 38.30 | 41.47 | 43.42 | 52.21 |
| RIGNN (Ours) | 60.13 | 66.41 | 63.56 | 65.56 | 69.70 | 78.23 | 43.73 | 47.10 | 36.38 | 37.74 | 50.14 | 45.05 | 55.31 |
| Improvement | 1.3% | 5.5% | -0.7% | -0.7% | 19.1% | 7.1% | 10.1% | 12.1% | -3.6% | -1.5% | 20.9% | 3.8% | 5.9% |

unlabeled graphs, which could belong to unknown classes. RIGNN has good generalization ability, and in the following, we would see that the model does well in open-set graph classification.

## 5.3 The Performance of RIGNN in Open-set Graph Classification (RQ2)

The performance of our RIGNN on the COIL-DEL, Letter-High, MNIST ,CIFAR10, REDDIT and COLORS-3 in comparison with several baseline methods is listed in Table 2. According to the results, we can see that the extended model generalizes well into the open-set graph classification task and outperforms all the listed baselines in both Hard and Easy scenarios on all four datasets. Similar to the open-world classification setting, the model gains a relative improvement of about 6.4% on average. The high performance in the open-set graph classification task shows that the extended model RIGNN is also good at detecting out-of-distribution instances, i.e., instances in unknown classes.

Furthermore, we can observe that although semi-supervised graph classification methods generally outperform the other GNN-based baselines, the extended RIGNN gains more improvement. This suggests that existing semi-supervised graph classification methods (*e.g.* GraphCL (You et al., 2020a) and GLA (Yue et al., 2022a)) are able to detect out-of-distribution instances more effectively than vanilla GNNs, they do so with lower accuracy than our proposed RIGNN and more importantly, they are weak in clustering the instances in the unknown classes into reasonable clusters, as can be seen from their performance in the open-world classification task.

## 5.4 Ablation Study (RQ3)

In this part, extensive ablated studies are conducted on the COIL-DEL and MNIST datasets to demonstrate the effectiveness of the proposed RIGNN. Concretely, we perform the experiments in the open-set graph classification setting and remove some of the proposed modules/mechanisms to test the prediction accuracy. The three variants of RIGNN include: (1) RIGNN w/o CRL, which removes the rationale representation learning and utilizes a single message passing neural network to generate graph representations; (2) RIGNN w/o D, which removes the disentanglement between rationale features and non-rationale features; (3) RIGNN w/o G, which removes the enhancement from the graph-of-graph. (4) RIGNN w CLUE, which employs a different popular estimator CLUE [2]. CLUB firstly models the distribution $q_\gamma(z^{nr}|z_r)$, and then calculates the mutual information using $I(Z^r, Z^{nr}) = \mathbb{E}_{\mathcal{G}_i \in \mathcal{D}}\left[\log q(z_i^{nr} \mid z_i^r)\right] - \mathbb{E}_{\mathcal{G}_i, \mathcal{G}_j \in \mathcal{D}}\left[\log q(z_j^{nr} \mid z_i^r)\right]$.

The results are summarized in Table 3. From the results, we have the following observations: (1) It is evident that removing each component causes the performance to drop in all cases, which demonstrates the contribution of rationale representation learning, disentanglement between rationale features and non-rationale features, and graph-of-graph representation enhancement. (2) The model experiences more significant performance drops when the rationale representation learning module is removed (*e.g.* 7.18% absolute percentage drop in COIL-DEL Hard task and 10.60% absolute percentage drop in MNIST Hard task). This suggests

Table 3: Ablation studies of our proposed RIGNN on the COIL-DEL and MNIST datasets. RIGNN w/o CRL removes the causal representation learning and utilizes a single message passing neural network to generate graph representations; RIGNN w/o D removes the disentanglement between causal features and non-causal features; RIGNN w/o G removes the enhancement from the graph-of-graph.

| Experiment | COIL-DEL | | MNIST | |
|---|---|---|---|---|
| | Hard | Easy | Hard | Easy |
| RIGNN w/o CRL | 52.95 | 59.74 | 59.10 | 65.32 |
| RIGNN w/o D | 56.15 | 63.21 | 65.04 | 75.92 |
| RIGNN w/o G | 57.44 | 62.56 | 63.39 | 74.86 |
| RIGNN w CLUE | 57.38 | 64.97 | 68.59 | 77.71 |
| RIGNN | 60.13 | 66.41 | 69.70 | 78.23 |

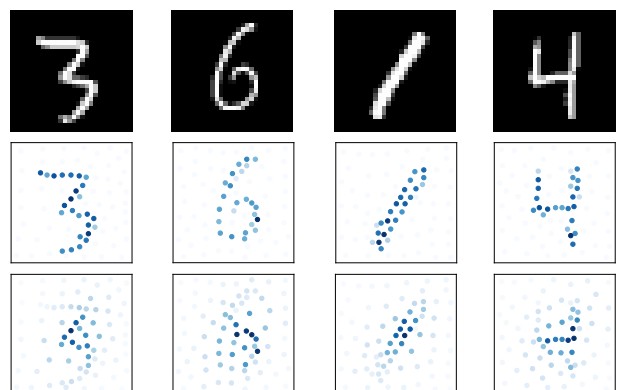

Figure 3: Visualization of learned important scores generated by the relational detector (Top line: original images; middle line: superpixel-based graphs; Bottom line: important scores). The experiments are performed on dataset MNIST, and darker nodes are relatively more important. The results show that the rationale detector in the proposed RIGNN is able to make a reasonable estimation of node importance.

that detecting rationale subgraphs in the original graph and removing non-rationale components is important for the performance in the face of unknown classes. (3) The use of contrastive learning in both rationale and graph-of-graph proximity contexts is helpful for the classification, since it can learn robust representations for the rationale part of the graph. This is in alignment with the results in the table: removing either rationale contrastive learning or the graph-of-graph construction hurts the prediction accuracy. (4) We can observe that the performance of RIGNN w CLUE remains relatively comparable to our original model. This suggests that the choice of MI estimator does not significantly impact the performance of our model. Nevertheless, we agree that it is important to select the appropriate MI estimator based on the characteristics of the data and the specific application. Our approach allows for flexibility in choosing the MI estimator based on validation data when dealing with new datasets.

## 5.5 Visualization (RQ4)

In addition, we offer some visualization results to show the effectiveness of the rationale representation learning in RIGNN. Concretely, we conduct experiments on the MNIST dataset and visualize the rationale-based important scores generated by the relational detector. We show the results in Figure 3. As can be observed from the results, the proposed rationale importance estimation in representation learning yields reasonable important scores related to their underlying patterns, which validates that our exploration of rationale factors can obtain meaningful subgraphs and thus learn effective graph representations. Moreover, the important

nodes tend to come together and therefore the selected rationale subgraph tends to be connected. In contrast, the nodes in the boundary tend to not be selected.

## 6   Discussion

In this part, we provide a discussion between our rationale-informed framework and causality-based graph representation methods. Causality has been widely utilized to learn graph-level representations in out-of-distribution generalization (Chen et al., 2022b; Sui et al., 2022), which identifies the causal mechanisms in graph generation and enhances the representation learning using the intervention. In our setting, we utilize mutual information maximization to discover the rationale elements in graphs without the intervention procedure, which is more close to rationale discovery instead of causality learning. However, we believe that causality learning is still valuable in graph representation learning with the invariance principle, and would explore the strength in the complex scenarios with out-of-distribution shift.

## 7   Conclusion

This paper studies the problem of semi-supervised open-world graph classification and a novel method named RIGNN is proposed to solve the problem, which detects features that hold the most information about the label space. Our RIGNN contains a relational detector and a feature extractor to provide rationale features. To capture rationale components, we maximize their mutual information with label information and require sufficient disentanglement with non-rationale components. In addition, we build a graph-of-graph based on geometrical relationships that provide guidance on improving rationale representations. We also make an extension for effective open-set graph classification. Interestingly, our method contains both mutual information maximization and minimization and we will investigate more about the relationship between them for enhanced performance. Comprehensive experiments on four popular datasets evaluate the efficacy of our proposed RIGNN. We will also test our model on wild datasets for a more comprehensive evaluation of its robustness and versatility.

## Funding Information

This work was partially supported by NSF 2211557, NSF 1937599, NSF 2119643, NSF 2303037, NSF 2312501, NASA, SRC, Okawa Foundation Grant, Amazon Research Awards, Cisco research grant, Picsart Gifts, and Snapchat Gifts.

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

## A  Algorithm

The algorithm of our RIGNN is summarized as below.

---
**Algorithm 1:** Training Algorithm of RIGNN

---
**Require:** Training set $\mathcal{D} = \mathcal{D}^l \cup \mathcal{D}^u$, parameter $\lambda$;
**Ensure:** The prediction for all unlabeled graphs;
 1: Initialize parameters $\theta$, $\phi$ and $\gamma$.
 2: **for** $t = 1, 2, \cdots, T$ **do**
 3:    Calculate the similarity and calculate $\mathcal{A}$ using Equation 14;
 4:    Update the positive set using Equation 15;
 5:    **repeat**
 6:      Generate a batch by sampling graph examples from $\mathcal{D}^l$ and $\mathcal{D}^u$;
 7:      Produce rationale features and non-rationale features using Equations 5 and 6;
 8:      Compute the overall loss with Equation 18;
 9:      Update the parameters in the network through back propagation;
10:    **until** convergence
11: **end for**

---

## B  Details of Baselines

Their details of the compared baselines are introduced as follows:

- Weisfeiler-Lehman (WL) Kernel (Shervashidze et al., 2011), which adopts the Weisfeiler-Lehman algorithm to construct a mapping from the original graph to a graph sequence.

- Shortest-Path (SP) Kernel (Borgwardt & Kriegel, 2005), which attempts to decompose graphs into various shortest paths for comparison.

- Graphlet Kernel (Shervashidze et al., 2009), which calculates the number of graphlets in the input graphs to generate features.

- GCN (Kipf & Welling, 2017), which is the pioneer graph neural network method. It to adopt the normalized adjacent matrix for message passing.

- GraphSAGE (Hamilton et al., 2017), which introduces sampling into efficient message propagation.

- GIN (Xu et al., 2019), which relates the power of message passing neural networks to the Weisfeiler-Lehman test.

- SAG Pooling (Lee et al., 2019a), which utilizes the attention mechanism to generate hierarchical subgraphs, which can generate effective graph representations for downstream tasks.

- GraphCL (You et al., 2020b), which introduces four graph augmentation strategies to compare different views, and can be extended to a semi-supervised graph classification method.

- GLC (Yue et al., 2022a), which utilizes label-invariant augmentation to enhance graph classification and tests the performance for semi-supervised graph classification.

## C  Hyperparameter Analysis

In this part, we study the parameter sensitivity in our proposed RIGNN. More specifically, we conduct experiments on the COIL-DEL and MNIST datasets for open-set graph classification. The results are shown in Figure 4. The first column shows the performance of the model as $k$ changes, while the second column

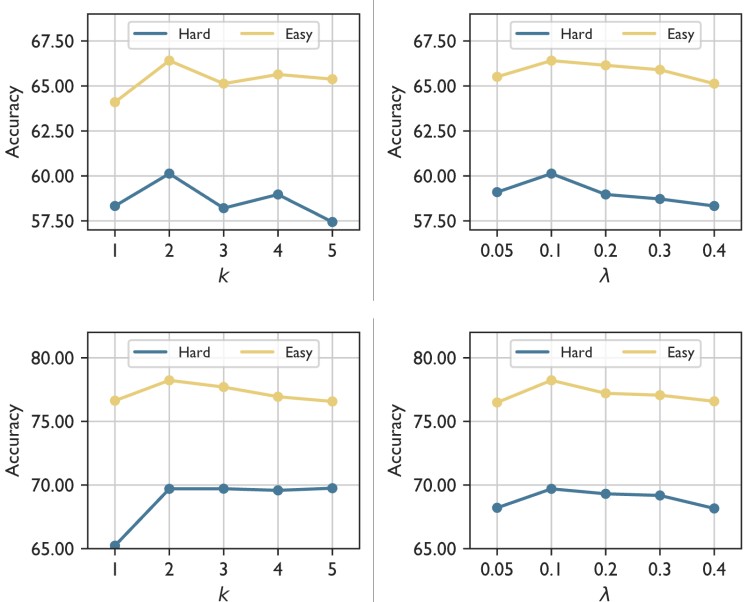

Figure 4: The parameters sensitivity analysis of our RIGNN for the open-set graph classification task, and we provide the result on both Easy and Hard scenarios. The top row shows the experiments on the COIL-DEL dataset, while the bottom row presents the experiments on the MNIST dataset.

shows the prediction accuracy as $\lambda$ changes. The upper part of the figure presents the results on the COIL-DEL dataset and the lower part shows the results on the MNIST dataset.

As can be seen from the results, the model is generally not sensitive to these hyperparameters and perturbing them in a specific range has limited influence on the classification accuracy. For hyperparameter $k$, we find that the approach obtains the best performance when it is set to 2. Decreasing $k$ will result in a relatively sparse graph and fewer anchors for the representation enhancement via graph-of-graph, whereas increasing $k$ will add noise to the contrastive objective, hurting the performance. Similarly, we find that our RIGNN has the best performance when $\lambda$ is set to 0.1, which provides the appropriate weight for the contrastive learning loss.

## D   Details of Datasets

The details of the adopted datasets are introduced as follows:

- *COIL-DEL.* The COIL-DEL dataset (Riesen & Bunke, 2008) is created by Harris corner detection as well as Delaunay Triangulation on image data. Then, a graph is constructed with nodes representing ending points and edges representing lines.

- *Letter-high.* The Letter-high dataset (Riesen & Bunke, 2008) is made of graphs indicating fifteen capital letters, i.e., A, E, F, H, I, K, L, M, N, T, V, W, X, Y, Z. In each graph sample, a node denotes an endpoint, and edges denote lines. Highly distorted letters indicate the high difficulty in identifying them.

- *MNIST.* The MNIST dataset (Dwivedi et al., 2020) is adapted from a vision dataset with the same name, where we extract super-pixels of images to construct nodes and a kNN graph is utilized to characterize the relationships between super-pixels.

- *CIFAR10.* The CIFAR10 dataset (Dwivedi et al., 2020) is also a vision dataset with a similar construction manner. Moreover, CIFAR10 is more challenging since it is made up of larger graphs with complicated semantic information.

Table 4: Statistics of the datasets used in the experiments.

| Dataset | # Graphs | # Classes | # Known | # Unknown |
|---|---|---|---|---|
| COIL-DEL | 3900 | 100 | 80 | 20 |
| Letter-High | 2250 | 15 | 10 | 5 |
| MNIST | 55,000 | 10 | 7 | 3 |
| CIFAR10 | 45,000 | 10 | 7 | 3 |
| REDDIT | 11929 | 11 | 7 | 4 |
| COLORS-3 | 10500 | 11 | 7 | 4 |

# E    More Visualization

We present the visualization result of the classification of the MNIST dataset. We compare our prediction with the prediction of GIN, and the result is listed in Figure 5. As can be seen from the results, the proposed RIGNN achieves better performance when there are unlabeled out-of-distribution data. The baseline model classifies the OOD instances into known classes, while our method detects the instances as out-of-distribution.

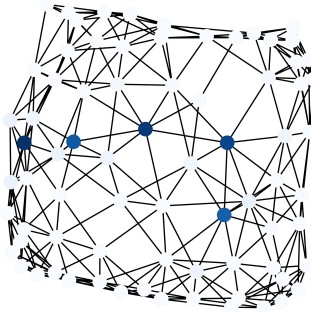 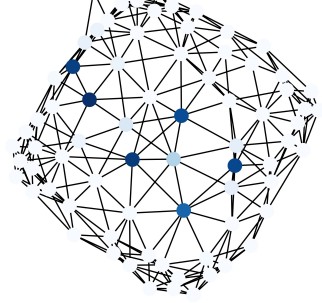

Ground Truth: 7
Baseline Prediction 1 (Wrong)
Our Prediction: OOD (Right)

Ground Truth: 9
Baseline Prediction 1 (Wrong)
Our Prediction: OOD (Right)

Figure 5: Visualization of two graph examples from the MNIST dataset. Our RIGNN can make the correct prediction while the baseline GIN cannot detect these out-of-distribution samples.

