# OpenReview forum: "RIGNN: A Rationale Perspective for Semi-supervised Open-world Graph Classification"
_TMLR — Accepted by TMLR_

### Review · Reviewer_4gxE · 2023-06-23

**Summary Of Contributions:**

The authors study the problem of semi-supervised open-world graph classification problem, where some novel classes are unseen during the training stage. The authors propose Casualty Informed GNN (CIGNN) to address the problem. Experimental results demonstrate the good empirical performance of CIGNN.

**Audience:**

Yes

**Broader Impact Concerns:**

I do not aware of broader impact concerns.

**Claims And Evidence:**

No

**Requested Changes:**

1. Improve the clarity of the paper. In particular address my questions above.

2. Further polish the paper. Some parts read really weird (i.e., those I mentioned above) and I feel polishing the paper will also improve the readability of the manuscript.


**Strengths And Weaknesses:**

## Strengths

(+) The open-world graph classification problem is interesting.

(+) Good empirical performance.

## Weaknesses

(-) The clarity of the paper should be greatly improved.

(-) The methodology seems not convincing.

(-) The writing should be further polished.

## Detail comments

I agree that the problem studied in this paper is interesting and may be important in practical scenarios. The proposed CIGNN method also demonstrates good empirical performance against baseline methods. However, I personally find the paper hard to digest and I am not convinced by the proposed methodology. Firstly, I am not an expert in causal analysis, so I might miss something about the causal analysis part of the paper.

According to my understanding, the goal of the paper is trying to learn two types of graph embedding, causal part $z^c$ and non-causal part $z^{nc}$. The high-level idea is that the downstream label $y$ is only determined by $z^c$ but not $z^{nc}$. This seems to be the design logic of equation 7. However, the logic to learn $z^c$ and $z^{nc}$ seems not unclear to me. I am not sure why using the approach defined in equation (4)-(6) is reasonable.

The description above equation (8) is also very unclear to me. The authors said that ``we consider two different graphs with an identical causal part, i.e., $\mathcal{G}$ and $\tilde{\mathcal{G}}$. From our SCM, their labels, i.e., $y$ and $\tilde{y}$ should be the same…’’. The message I got here is that if two graphs have the same causal embedding $z^c = \tilde{z}^c$, then their label would be the same. However, equation (8) says we should maximize the mutual information between $ z^c,\tilde{z}^c$ given $y=\tilde{y}$. This seems not logically correct unless the former statement (in the quote) represents an if and only if relation. That is, $ z^c = \tilde{z}^c \Rightarrow y=\tilde{y}$ does not imply $ z^c = \tilde{z}^c \Leftarrow y=\tilde{y}$. I feel this part needs more explanation and to be more precise.

The way to tackle the unlabeled graph in the contrastive learning part is also not clear. How to choose the subgraph of the unlabeled graph? Why the subgraph embedding should be similar to the original graph? I feel many details are missing here. Finally, I do not understand the relationship between the contrastive learning and causal learning part of the objective. Does the author include $\mathcal{L}_{con}$ merely to leverage the unlabeled graphs in some sense? What is the relation between these two parts?

Some parts read weirdly in the paper:

1. Is there a typo in equation (3)? The denominator seems weird. The summation should enumerate over $i^\prime$ but the authors wrote $\sum_{G_{i^\prime}\in B}$, where $G_{i^\prime}$ is not defined.

2. Section 4, second paragraph: “causal-attended” should be “causality-informed”? Otherwise why the authors named it CIGNN?

3. Section 4: “…to make use of abundant labeled graphs…” I thought the authors claim the label is scarce in the introduction?

---

### Review · Reviewer_HeaJ · 2023-06-25

**Summary Of Contributions:**

The paper proposes a novel GNN called Causality-Informed GNN (CIGNN) for semi-supervised open-world graph classification, where unlabeled graph data could come from unseen classes. CIGNN takes a causal look to detect components containing the most information related to the label space and classify unlabeled graphs into a known class or an unseen class. In particular, CIGNN contains a relational detector and a feature extractor to produce effective causal features, which maximize the mutual information with label information and exhibit sufficient disentanglement with non-causal elements. Furthermore, the authors construct a graph-of-graph based on geometrical relationships, which gives instructions on enhancing causal representations. In virtue of effective causal representations, they can provide accurate and balanced predictions for unlabeled graphs. The proposed method is verified on four benchmark datasets in various settings, and experimental results reveal the effectiveness of CIGNN compared with state-of-the-art methods.

**Audience:**

Yes

**Broader Impact Concerns:**

I don't have any broader impact concerns.

**Claims And Evidence:**

No

**Requested Changes:**

Please see **Weaknesses** above.

**Strengths And Weaknesses:**

## Strengths:

**1.** CIGNN has the ability to effectively extract causal features (w.r.t. semantic labels) from graphs while reducing the impact of non-causal components. With its use of a relational detector and feature extractor, the framework can produce effective causal features, and the authors also propose a construction of a graph-of-graph based on geometrical relationships to enhance and visualize causal representations.

**2.** Additionally, different from traditional semi-supervised learning, CIGNN can provide accurate and balanced predictions for unlabeled graphs that are even outside the predefined label class, even in the presence of label scarcity.

**3.** The paper is well-structured and experimental results reveal the effectiveness of CIGNN compared with state-of-the-art methods on various benchmark datasets in various settings that may have potential implications for the causal and graph machine learning.

## Weaknesses:

### Major Comments:

However, the reviewer may have the following concern regarding this paper:

**1.** Regarding the SCM in Section 4.2: in literature such as Pearl (2009) and Peters et al. (2017), SCM is a framework used in causal inference and modeling to understand the relationships between variables and their causal dependencies. For example, given a DAG, we would recognize that the variable X causes variable Y if X pointed out to Y. But in figure 2, the author does not really explain the definition of the causality and non-causality when first introducing it, e.g., what does causal part feature mean in the context, with respect to which variables? Please make the motivation clearer so that readers would be convinced that the method is well-based on causality. If you are referring to causality as w.r.t. the semantic label Y at a later stage, should we exclude those features as theoretically they are non-causal instead of reducing the impact of NC? Otherwise, I would only recognize that Figure 2 as a high-level idea of finding useful (or more impactful) features for predictions.

**2.** Many notations are not properly defined and used, which makes the paper less readable, please refer to minor coments for more details. Further, going through the whole paper, I feel that using the mutual information does not go beyond statistics. Say, maximizing the mutual information does not necessarily mean doing causal discovery or finding causal features. I would say this is a process of finding label-relevant/irrelevant representations from graph and it basically does not leverage rigorous causal definitions or methods such as finding the causal relationship between two nodes, or investigating the graph structure with ideas of interventions.

**3.** While I like figure 4 that the author can extract and visualize the causal features but it is not necessarily helpful in its current form as readers could not see useful information from its current graph presentation. Instead, I would suggest enhancing the visual representation by highlighting the key pixels (or nodes) in real images based on these causal features. This approach will provide readers with a better understanding of the performance of CIGNN during predictions.

### Minor comments:

**1**. In (1), what is $k$ in your notation? Do you mean $l-1$? In (2), does $L$ necessarily mean the total layer number? Please specify these notations clearly.

**2**. What does $\mathbb{E}$ mean in (3)? Does it mean "empirical average" over dataset? While typically it means the expectation over a random variable.

**3**. In (4.1), typo in "…, we propose a novel method named causal-attended graph neural network (CIGNN)…", should it be causal-informed as the abstract tells? While in the later context, authors are all using "causal-attended" instead of "causal-informed". Please make them consistent.

**4**. “We first present a structure causal model (SCM)” -> structural causal model.

**5**. In (4), what does $f_\theta(i;\mathcal{G})$ indicate without appropriately defining it? Do you mean the ith vertex of $f_\theta(\mathcal{G})$? Further, what is $\alpha$ in “..retain $\alpha|\mathcal{V}|$ nodes.”?

**6**. Define or mention Hadamard product when using $\mathbf{X} \odot \mathbf{M}$.

**7**. Throughout the whole paper, the mutual information between $X$ and $Y$ should be written as $I(X;Y)$, instead of $I(X,Y)$ with the comma.

**8**. Eq (8) is not mathematically right – please refer to the definition of conditional mutual information such as Cover & Thomas (2006). I could understand (7) if the authors are referring a certain graph as a random variable (and its label $y$ as well), and maximization of mutual information between $z^c$ and $y$ means the learned random feature $z^c$ contains high information of the random label $y$. But when it comes to conditional mutual information, where do $\tilde{z}^c$ and $\tilde{G}$ come from? Is $\tilde{G}$ also a random graph and does $\tilde{G}$ have the same distribution as $G$? Then your interpretation might not be very appropriate as you explain that from an individual graph level. Apart from that, conditioning on $y = \tilde{y}$ is not properly used in the information theory: a better notation would probably be something like $I(z^c_Y ; \tilde{z}^c_Y |Y)$ where $z_Y^c$ and $\tilde{z}_Y^c$ corresponds to the feature representation of $G$ and $\tilde{G}$ given the same label $Y$.

**9**. What is $\epsilon$ after (9)? Might be a typo for $\tau$.

**10**. In (16), what does $\phi_\phi$ mean?  Might be a typo for $h_\phi$?

**11**. In experimental section, what are the label proportions of **Easy** and **Hard** Problems? Please specify the details.

References:
1. Pearl, J. (2009). Causal inference in statistics: An overview.
2. Peters, J., Janzing, D., & Schölkopf, B. (2017). Elements of causal inference: foundations and learning algorithms (p. 288). The MIT Press.
3. Thomas, M., & Joy, A. T. (2006). Elements of information theory.

---

### Review · Reviewer_ecKj · 2023-06-30

**Summary Of Contributions:**

The paper tackles the open-world semi-supervised graph classification where unlabeled graphs could come from unseen classes. The authors propose CIGNN, which takes a causal perspective to detect components containing the most information related to the label space and classify unlabeled graphs into known or unseen classes. The approach includes a relational detector and a feature extractor to produce causal features. In addition, a graph-of-graph is constructed to enhance causal representations. The paper also mentions the extension of the approach to open-set graph classification. Experimental results show the effectiveness of CIGNN compared to baseline methods such as GCN, UGNN.

**Audience:**

Yes

**Broader Impact Concerns:**

There are no significant broader impact concerns for this paper.

**Claims And Evidence:**

Yes

**Requested Changes:**

1. Please add the details of the training to address the training stability questions, and add potential discussions regarding the effects of $\gamma$.

2. Please add results on at least REDDIT-MULTI-12K and COLORS-3 datasets if possible.

**Strengths And Weaknesses:**

## Strengths

The paper is overall well-written and organized, making it easy to follow. I did not find significant grammatical errors or clarity concerns. The proposed loss is well-motivated. From the empirical perspective, the authors have shown the proposed method has better performance on open-set classification and OOD in various datasets. Some visualization results of the causal representation are also provided in the paper.

## Weaknesses

Most of my concerns fall in the discussion in terms of the mutual information. The authors propose to tackle the non-causal impact with the maximization of mutual information. However, several key perspectives regarding mutual information estimation are not complete:

The loss of disentanglement $\mathcal L_{dis}$ is dependent to a mutual information estimator $T_\gamma$ that is trained in an adversarial manner. The training stability might be a concern of the proposed CIGNN, as the scale of all components in Eq (18) is different. The authors may need to provide more details like papers training GANs in addressing the stability in training. For example, do we need separate optimizers, do $\theta$, $\phi$ and $\gamma$ take different update frequencies, etc.

The mutual information estimators are usually upper-bound or lower-bound estimators. Some of them are unbiased but have high variance, while some are biased but have lower variance. Different estimators have different properties and request different sample selections to get a good approximation. How the quality of the estimator affects CIGNN is not theoretically analyzed nor empirically studied.

The mutual information estimators are sometimes sensitive to the number of samples used in the estimation. The authors may also need to conduct corresponding studies to address this point.

A complementary point regarding the mutual information estimators is the relation between $\mathcal L_{dis}$ and $\mathcal L_{con}$, since JS-bound estimator and contrastive prediction estimator are two different estimators. It would be interesting if the authors could explore deeper on the relation between this two components. (Please note we do not request corresponding changes regarding this point.)


Another of my concern is the comparison with the baselines in Table 3. Several of the baselines also report the results on REDDIT-MULTI-12K and COLORS-3 datasets. Are there any specific reasons why the authors eliminate these two datasets?

As the paper tackles the open-set  problems, it could be more convincing if the authors could provide experiment results on some real-world datasets. E.g., images in the wild data.

---

### Public Comment · ~Longfei_Ma1 · 2023-12-20
**Code for reproduction**

Dear Authors,

I find your work to be fascinating. I am particularly interested in delving deeper into the implementation aspects of your research.
Could you please provide information on how I could access the code associated with your paper? I believe it would greatly enhance my understanding and potentially contribute to my own research endeavors.

Thank you for your time and consideration.

Best regards,

---

### Decision · Action_Editors · 2023-08-02

**Recommendation:** Accept with minor revision

**Comment:**

This paper proposes a novel approach for open-world graph classification, where the unlabeled graph may come from classes that do not belong to the training label space. The proposed method was developed from a causal perspective, which detects components containing the most information related to the label space and classify unlabeled graphs into known or unseen classes. However, the reviewers pointed out the method does not fully explore the properties of a causal model and thus one may not need to go to the causal level to develop the proposed method. Thanks to the reviewers' comments, the authors have made substantial modifications to remove the causal inspirations and introduced a new "rationale" perspective. After this significant modification and additional experiments, I think the current version still presents a solid technical contribution, but with more sensible motivations. I would recommend acceptance of the paper contingent on addressing the following revisions:

1. Ensure consistency in presenting the new "rationale" perspective throughout the paper.
2. Review the entire manuscript to verify that the updated approach is well-integrated and clearly explained.
3. Discuss why the causal perspective may not be necessary in the proposed method and whether causality still have a potential in developing new and better methods in the future.




**Audience:**

Yes

**Claims And Evidence:**

Yes